

# Identification, validation and quantification of thymoquinone in conjunction with assessment of bioactive possessions and GC-MS profiling of pharmaceutically valuable crop Nigella (*Nigella sativa* L.) varieties

Ravi Y[1,2], Irene Vethamoni Periyanadar[2], Shailendra Nath Saxena[1], Raveendran Muthurajan[3], Velmurugan Sundararajan[2], Santhanakrishnan Vichangal Pridiuldi[2], Sumer Singh Meena[1], Ashoka Narayana Naik[4], C. B. Harisha[5], Honnappa Asangi[6], Sharda Choudhary[1], Ravindra Singh[1], Yallappa Dengeru[7], Kavan Kumar V[8], Narottam Kumar Meena[1], Ram Swaroop Meena[1] and Arvind Kumar Verma[1]

[1] ICAR-National Research Centre on Seed Spices, Ajmer, Rajasthan, India
[2] HC & RI, Tamil Nadu Agricultural University, Coimbatore, India
[3] Centre for Plant Molecular Biology and Biotechnology, Tamil Nadu Agricultural University, Coimbatore, Tamil Nadu, India
[4] COH, Sirsi, University of Horticultural Sciences, Bagalkote, Sirsi, Karnataka, India
[5] ICAR-National Institute of Abiotic Stress Management, Pune, Maharastra, India
[6] ICAR-Indian Institute of Spices Research, Regional Station, Appangala, Madikeri, Karnataka, India
[7] AEC & RI, Tamil Nadu Agricultural University, Coimbatore, India
[8] Department of Renewable Energy Engineering, CTAE, MPUAT, Udaipur, India

Corresponding author
Ravi Y, ry.davanagere@gmail.com

## ABSTRACT

**Background:** Plants have been pivotal in traditional and modern medicine globally, with historical evidence supporting their therapeutic applications. Nigella (*Nigella sativa* L.) is an annual herbaceous plant of the Ranunculaceae family and is cultivated in the Middle East, Eastern Europe, and Western and Central Asia. The medicinal use of plants dates back thousands of years, documented in ancient writings from various civilizations. Alkaloids, phenolics, saponins, flavonoids, terpenoids, anthraquinones, and tannins found in plants exhibit antioxidant, immunomodulatory, anti-inflammatory, anticancer, antibacterial, and antidiabetic activities.

**Methodology:** This study specifically examines the pharmacological potential of *Nigella sativa* L., emphasizing thymoquinone—a compound with diverse nutraceutical benefits. The extraction, characterization, and quantification of thymoquinone, alongside other physicochemical parameters, were carried out using ethanol through Soxhlet extraction procedures on five nigella varieties. HPLC analysis was performed to determine the maximum accumulation of thymoquinone
in the released variety of the plant and the chemical composition of the seed oil isolated from *Nigella sativa* L., varieties utilized in the study was determined through GC-MS analysis.

**Results:** The research revealed that the Ajmer nigella-20 variety stands out, exhibiting elevated levels of thymoquinone ($0.20 \pm 0.07\%$), antioxidants ($76.18 \pm 1.78\%$), and substantial quantities of total phenols ($31.85 \pm 0.97$ mg GAEg$^{-1}$ seed) and flavonoids ($8.150 \pm 0.360$ mg QE 100 g$^{-1}$ seed) compared to other varieties. The GC-MS profiling showed the presence of 11 major compounds in the studied varieties, with p-cymene, longifolene, and myristic acid identified as the major chemical compounds present in the oil.

**Conclusion:** The observed variations among Nigella varieties indicate the Ajmer nigella-20 variety as particularly promising for thymoquinone and bioactive compound extraction. This study underscores Nigella's potential as a source of pharmacologically active compounds, highlighting the need for further exploration in therapeutic applications.

# INTRODUCTION

*Nigella sativa* L., commonly known as black seed or Kalounji, is an annual herbaceous plant belonging to the Ranunculaceae family, originating from the Mediterranean region and thriving in Eastern Europe, the Middle East, and Western Asia (*Ravi et al., 2022*). The yellowish volatile oil and fixed oil derived from its seeds are abundant sources of metabolites, including proteins, amino acids, reducing sugars, mucilage, alkaloids, organic acids, tannins, resins, metarbin, glycosidal saponins, crude fiber, minerals, and vitamins (*Ramadan, Kroh & Mörsel, 2003*). Utilized for thousands of years as a spice and food preservative, nigella seeds have been recognized for their beneficial effects, encompassing antioxidant, anti-inflammatory, immunomodulatory, antimicrobial, anticancer, and antidiabetic properties (*Omojate Godstime et al., 2014*).

Notably, nigella exhibits a substantial amount of phenolic chemicals, rivalling oregano (211 to 270 mg GAE/g dried extract) (*Gutiérrez-Grijalva et al., 2017*) and thyme (99 to 208 mg TAE/g extract) (*Pirbalouti et al., 2014*), serving as rich sources of antioxidants (*Mariod et al., 2009*). Additionally, bioactive substances such as sterols, tocols, and essential fatty acids contribute to its therapeutic profile (*Piras et al., 2013*). The active metabolites isolated from nigella include thymoquinone (TQ), dithymoquinone, thymohydroquinone, carvacrol, p-cymene, t-anethol, terpineol, pinene, sesquiterpene longifolene, and thymol, with isoquinoline alkaloids and indazole or pyrazol alkaloids, including nigellicine and nigellidine, present in trace levels (*Kooti et al., 2016*; *Abdel-Moneim et al., 2013*).

Chemical structure of thymoquinone.

Thymoquinone (2-isopropyl-5 methyl-1,4-benzoquinone) a key active metabolite found in nigella oil extracts, has demonstrated antioxidant and anti-inflammatory properties in various *in-vitro* and *in-vivo* models, addressing conditions such as asthma, diabetes, encephalomyelitis, neurodegeneration, and carcinogenesis (*Ramadan, Kroh & Mörsel, 2003*; *Woo et al., 2012*). Furthermore, black seed extracts have exhibited antihistaminic, immune-potentiating, hypertensive, antibacterial, and anti-inflammatory properties, primarily attributed to active quinone-based components (*Mahfouz, 1960*; *Medinica et al., 1994*).

This study was designed to quantify TQ as an active constituent in nationally released varieties of *Nigella sativa* L., for commercial cultivation. Oil extraction methods used to extract aromatic oils from medicinal and aromatic plants play a major impact on the properties of oils and the quantification of the bioactive substances (*Mahendiran et al., 2023*). Soxhlet extraction/solvent extraction is the advanced essential oil distillation technique, where the solvent, selected for its affinity with the desired plant compounds, flows through the prepared biomass to penetrate the plant structures and release the essential oils. The resulting mixture of solvent, plant oils, and botanical solids is typically filtered and vacuum distilled to remove as much solvent as possible, especially when petroleum-based hydrocarbons are used (*Pokorný & Korczak, 2001*). The other methods of essential oil extraction from medicinal and aromatic plants includes hydro distillation, steam distillation and hydro-stem distillation. The drawbacks of these methods are long extraction time, possible chemical changes in the structures of terpenes, and the loss of some polar molecules owing to the applied heat. Therefore, the selection of extraction methods and solvent plays a significant role in isolating and identifying the bioactive molecules from the raw materials. In this study, we employed a Soxhlet extraction method to extract the total oil from nigella seeds and a rotary evaporator to evaporate the solvent. GC-MS profiling is the most widely used technique in metabolomics research to identify and characterise the structural characteristics of phytochemical compounds (*Jiang et al., 2020*). Considering that it is a very accurate method for characterising and identifying metabolites. The GC-MS is a useful tool for metabolite profiling because it is vital for the qualitative and quantitative characterization of chemical compound (metabolite) present in plant cells (*Cupido et al., 2022*). Genetic variability is an important factor for varying extents of metabolite obtained in different varieties of crop plants. The amount of active metabolite in various cultivars of the same plant growing in various parts of the country has been investigated in conjunction with a number of medicinal plants. The objective of

**Table 1 *Nigella sativa* L. varieties released for commercial cultivation in India.**

| Sl. No. | Variety | Characters |
|---|---|---|
| 1. | Ajmer Nigella-1 | Developed by ICAR-NRCSS, Ajmer. Plants are 30–35 cm tall, matures in 135 days, resistant to root rot, essential oil content is 0.3% and seed yield up to 8 q ha$^{-1}$ |
| 2. | Ajmer Nigella-20 | Developed by ICAR-NRCSS, Ajmer. Released during 2019–20, rich in oleic acid (3.32%), seed yield upto 9.09 q ha$^{-1}$. |
| 3. | Azad Kalonji | Developed by CSAUAT, Regional Research Station, Kalyanpur, Kanpur. Crop matures in 135–145 days, average seed yield is 900–1,000 kg ha$^{-1}$. |
| 4. | Pant Krishna | Developed by GBPUAT, Pantnagar. Seeds are medium in size and bold, appropriate for cultivation in Uttar Pradesh and Uttarakhand. |
| 5. | NDBC-10 | Developed from NDUAT, Ayodhya. Bears medium sized seeds and optimum yielder, suitable for cultivation under Uttar Pradesh condition |

**Note:**
NDBC-10, narendra deva black cumin-10.

this study was to identify, validate and quantify the thymoquinone metabolite from the released varieties of *Nigella sativa* L., *via* High Performance Liquid Chromatography (HPLC) method. Further, the Nigella plant has a potent therapeutic values, hence the authors explored the chromatographic heterogeneity (GC-MS) to estimate the bioactive compounds in the released verities.

# MATERIALS AND METHODS

## Chemicals

The standard thymoquinone and all solvents utilized were procured from Sigma–Aldrich (St. Louis, MO, USA), ensuring the highest available purity.

## Plant material

Seeds from nationally released varieties of *Nigella sativa* L., for commercial cultivation, along with their sources of collection and respective characteristics, are outlined in Table 1. The collected seed samples underwent a thorough washing in double-distilled water and were subsequently dried under ambient conditions. For further analysis, precisely weighed 50 g seed samples were ground into a powder using a Malavasi mill (Bologna, Italy). Particular attention was paid to prevent overheating during the grinding process, and the resulting particle sizes fell within the range of 250–425 µm.

## Extraction from Nigella seeds by Soxhlet apparatus

The extraction from seeds of various *Nigella sativa* L., varieties was performed using a Soxhlet apparatus. In this process, 15 g of powdered, dry seeds were immersed in 50 ml of absolute n-hexane at 65 °C for 6 h, incorporating minor modifications. Subsequently, the extract underwent rotary flash evaporation under reduced pressure to eliminate solvent residue. The resulting oily extract was then collected and stored under refrigerated conditions for further analysis. To quantify the efficiency of the extraction process, the extraction yield was calculated as a percentage of the raw material. This was achieved using the formula outlined by *Zhang, Bi & Liu (2007)*.

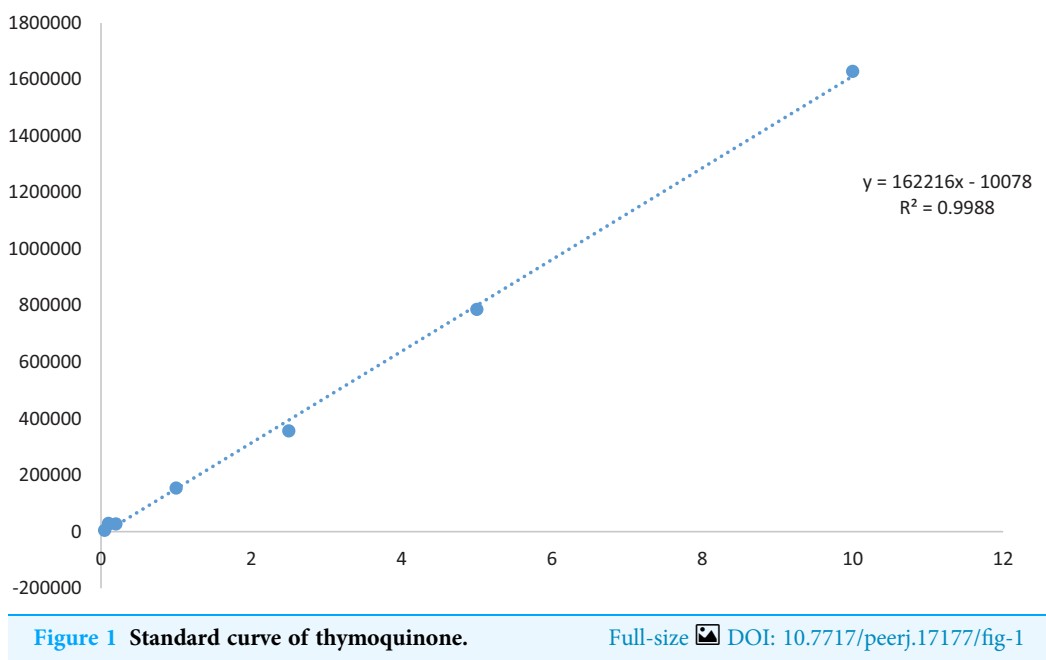

**Figure 1  Standard curve of thymoquinone.** 

$$\text{Extraction yield } (\%) = \frac{\text{Weight of the recovered sample}}{\text{Weight of the original sample}} \text{X } 100.$$

## Identification of thymoquinone *via* thin-layer chromatography

TLC plates (Merck, Darmstadt, Germany) characterized by silica gel 60G and dimensions of (4.5 mm × 10 mm) were employed for the identification of TQ. The extracted oil was spotted on the TLC plate and kept for development of the chromatograms. The development system was made up of *n*-hexane:ethyl acetate (8:2 v/v), which produced a strong and well-defined band for the metabolite and the identity was validated by comparing the bands of standard thymoquinone with those of studied sample extracts based on the visibility of bands under ultraviolet light.

## HPLC analysis

### Instrumentation and reagents

The entire oil obtained through Soxhlet extraction underwent high-performance liquid chromatography (HPLC) analysis using an Agilent 1,200 HPLC system equipped with a diode array detector (HPLC-DAD) (Agilent Technologies, Palo Alto, CA, USA). A C18 column (250 × 4.6 mm ID; 5 μm particle size, Agilent Technologies, Palo Alto, CA, USA) was employed with a mobile phase comprising water and methanol in a 40:60 ratio. Prior to use, the mobile phase was sonicated at 37 °C for 10 min, filtered through a 0.45 mm Millipore filter, and the injection volume was set at 20 μL. Thymoquinone was identified at a wavelength of 254 nm under ambient temperature. A flow rate of 1.5 mL/min was maintained, and identification was validated by comparing retention times. Both the standard thymoquinone and the sample content were present in the oil sample, and quantity calculations were performed using conventional linear calibration curves (*Hadad et al., 2014*).

## Standard calibration

A TQ stock solution with a concentration of 1.0 mg ml$^{-1}$ was prepared by dissolving 1.0 mg of TQ in HPLC-grade methanol, and the volume was adjusted to 1 ml. The solution underwent sonication at 37 °C for 10 min. Prior to HPLC analysis, all dilutions were meticulously made stepwise from the main stock in methanol, followed by filtration through a 0.22 µm membrane filter. Seven reference points for the TQ calibration curve (0.05, 0.1, 0.2, 1, 2.5, 5, 10 µg/mL) in methanol were established and are illustrated in Fig. 1. The calibration curves were generated using TQ (1.0 mg ml$^{-1}$) through linear least square regression, plotting the analysis concentration against the peak area. The lowest concentration on the calibration curve represents the lower limit of quantification (LLOQ), crucial for determining accurate and precise findings.

## GC-MS profiling

The chemical composition of the seed oil isolated from *Nigella sativa* L., varieties utilized in the study was determined through GC-MS analysis following the AOCS Method CE 1-62 and *Cupido et al. (2022)* with minor modifications. The GC-MS profiling was done using Agilent (GC-7820 A, MS-5975; Agilent Technologies, Santa Clara, CA, USA) equipped with an HP 5 (universal column; 30 m × 0.325 mm, film thickness 0.25 µm) Agilent J&W GC column with an autosampler. A sample of 1 ml was used with an autosampler. Helium was used as the carrier gas at a flow rate of 1.0 ml/min. The column temperature was programmed from 50 °C to 280 °C with an equilibrium time of 3 min, held for 30 min. Injector temperatures were set at 250 °C. The biochemical constituents were identified by a comparison of their retention indices and their identification was confirmed by computer matching of their mass spectral fragmentation patterns of compounds in the NIST-MS library and published mass spectra with the help of Chemtation software (Agilent Technologies, Palo Alto, CA, USA).

## Determination of total phenols

Twenty grams of dried seeds from each variety underwent sonication and extraction in 80 percent n-hexane, resulting in a total extract with a solid-to-solvent ratio of 1:10 (w/v). This extraction process was carried out at room temperature for 1 h, followed by thorough mixing. The concentrated solvent was removed using a rotary evaporator (Buchi, Flawil, Switzerland). To determine the total phenolic content (TPC) of the Nigella varieties, the Folin–Ciocalteu reagent and gallic acid as a standard were employed. A UV-Vis spectrophotometer (Shimadzu, Co., Ltd., Kyoto, Japan) was utilized to measure the absorbance at 760 nm. The TPC of the sample was then calculated as mg gallic acid equivalents (GAE) per 100 g of seed, following the method outlined by *Singleton & Rossi (1965)*.

## Total flavonoid content

The total flavonoid concentration of the extracts was determined using the aluminum chloride colorimetric method. In brief, 0.5 ml of the extract was mixed with 1.5 ml of distilled water and 0.5 ml of aluminum chloride (10% w/v). To this solution, 0.1 ml of potassium acetate (1 M) and 2.8 ml of distilled water were added. The reaction mixture was

**Table 2 Percentage yield, percent composition of thymoquinone and bioactive properties of *Nigella sativa* varieties.**

| Variety | Soxhlet extract | | Antioxidant activity (%) | Total phenols (mg GAE g$^{-1}$ seed) | Total flavonoids (mg QE 100 g$^{-1}$ seed) |
|---------|---------|---------------|--------------------------|--------------------------------------|---------------------------------------------|
| | % yield | % composition | | | |
| Ajmer Nigella-1 | 5.41 ± 0.16 | 0.13 ± 0.02 | 74.20 ± 2.64 | 31.15 ± 0.16 | 7.92 ± 0.34 |
| Ajmer Nigella-20 | 5.82 ± 0.03 | 0.20 ± 0.08 | 76.18 ± 1.78 | 31.85 ± 0.97 | 8.15 ± 0.36 |
| Azad Kalonji | 4.32 ± 0.03 | 0.11 ± 0.01 | 74.01 ± 0.26 | 31.23 ± 1.04 | 8.01 ± 0.28 |
| Pant Krishna | 4.98 ± 0.19 | 0.13 ± 0.06 | 72.18 ± 1.95 | 30.25 ± 1.11 | 7.80 ± 0.23 |
| NDBC-10 | 5.12 ± 0.12 | 0.12 ± 0.05 | 73.89 ± 1.79 | 30.18 ± 0.65 | 7.31 ± 0.11 |

**Note:**
% per cent; GAE, gallic acid equivalent; QE, quercetin quivalent; mg, mille gram; TQ, thymoquinone.

then stirred at room temperature for 30 min before measurement using a UV–Vis spectrophotometer at 415 nm wavelength (Shimadzu, Co., Ltd., Kyoto, Japan). The results were expressed as mg quercetin equivalent per 100 g of seed (mg QE 100 g$^{-1}$ seed), with flavonoid quantification done relative to quercetin as a standard.

## Antioxidant activity
### 1, 1-Diphenyl-2-picrylhydrazyl radical scavenging activity test
The antioxidant activity percentage of the phenolic isolate from Nigella varieties was assessed using the stable radical 1,1-diphenyl-2-picrylhydrazyl (DPPH). The procedure, as per *Brand-Williams, Cuvelier & Berset (1995)*, with minor modifications. The reaction mixture comprised 0.5 ml of the sample, 3.0 ml of absolute ethanol, and 0.3 ml of DPPH (0.5 mM). The reaction mixture was allowed to settle down at room temperature under dark condition for 60 min to complete the reaction. The decrease in the reaction was measured at 517 nm using a UV–VIS spectrophotometer (Shimadzu, Co., Ltd., Kyoto, Japan) and control solution was prepared by combining 3.5 mL of ethanol and 0.3 mL of DPPH. The scavenging activity was calculated following the formula outlined by *Gew et al. (2024)* and expressed as a percentage.

$$\text{Free radicle scavenging activity } (\%) = \frac{\text{Abs}_{control} - \text{Abs}_{sample}}{\text{Abs}_{control}} \times 100$$

## Statistical analysis
The mean and standard deviation were computed, and a graph was generated. Additionally, MS Excel (Microsoft, Redmond, WA, USA) was employed for ANOVA calculations.

# RESULTS AND DISCUSSION
## Percentage yield of extract in released varieties
Using *n*-hexane as the solvent, extracts of *Nigella sativa* L., were prepared from five distinct varieties to investigate and screen the most suitable plant type for thymoquinone concerning consumer value. As presented in Table 2, the percentage yields varied from 4.32% to 5.82%. The highest yield was observed in Ajmer Nigella-20 (5.82 ± 0.03%), while

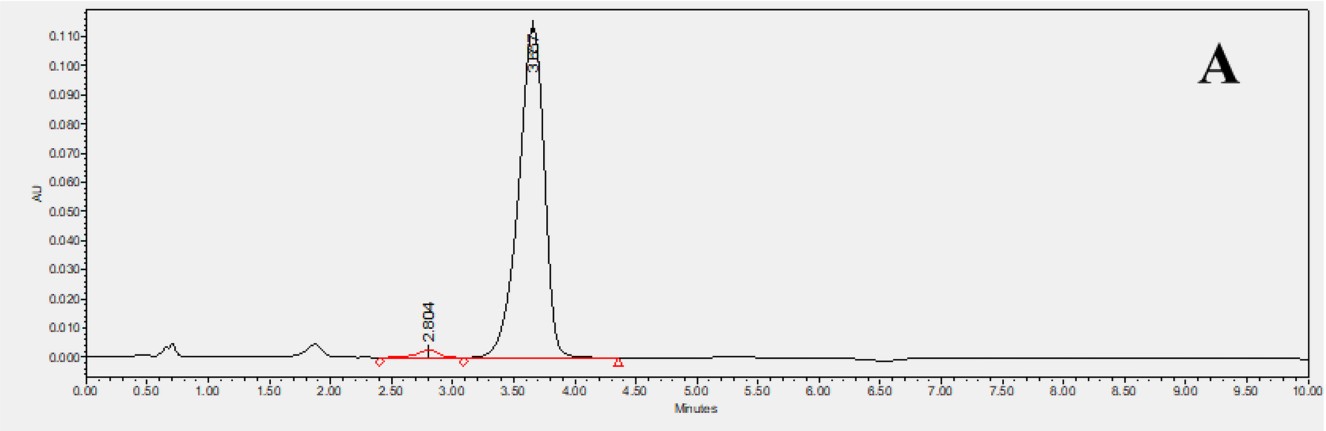

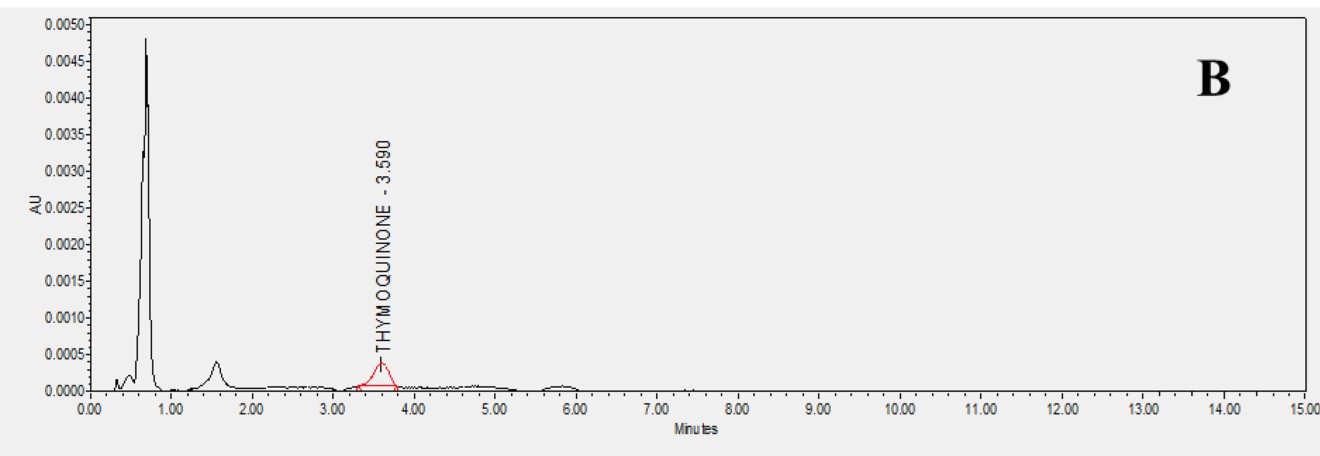

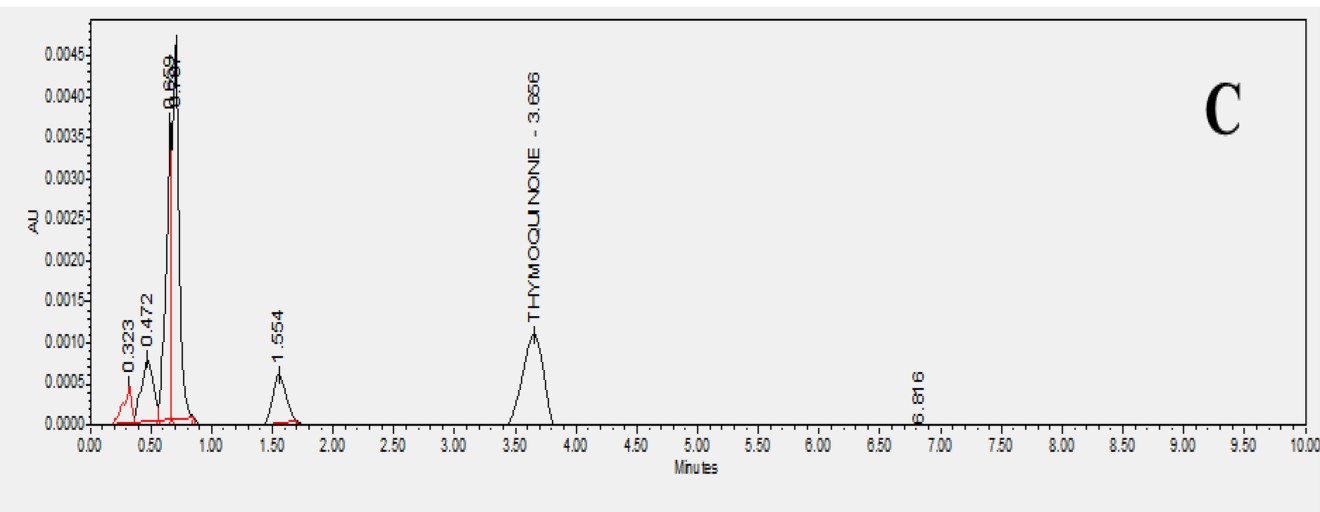

**Figure 2 HPLC chromatograms of standard and studied varieties.**

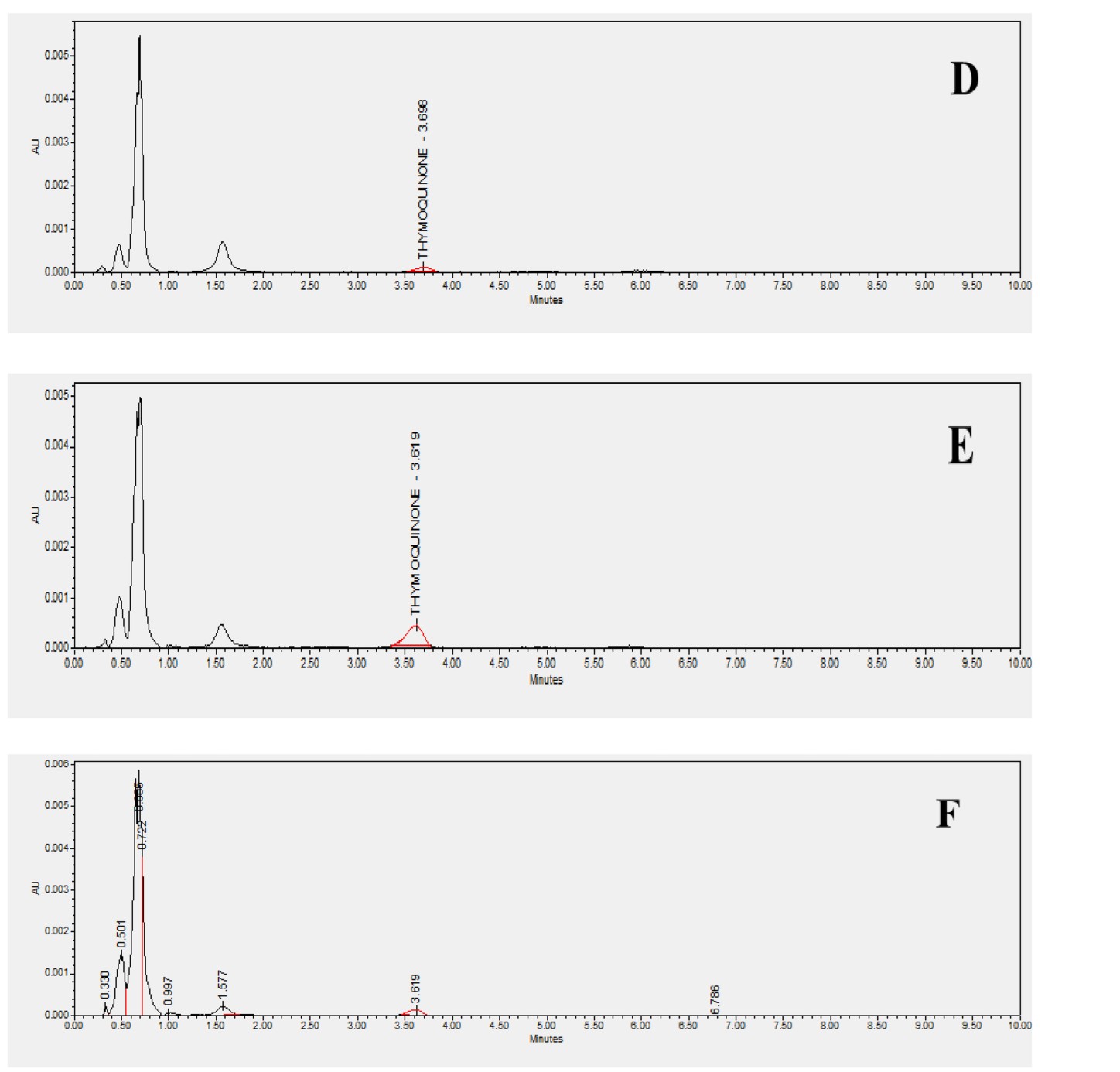

**Figure 2 HPLC chromatograms of standard and studied varieties. (Continued).**

the variety Azad kalonji exhibited the lowest yield (4.32 ± 0.03%). Similar results were seen in a study conducted by *Gupta et al. (2021)*, in which the methanol extract demonstrated an optimal yield. A yield study usually reveals a better foundation for bioactive compound extraction when more amounts of a particular solvent are used. Methanol extract was utilized to assess the quantity of thymoquinone in a specific variety of the plant, *Nigella*
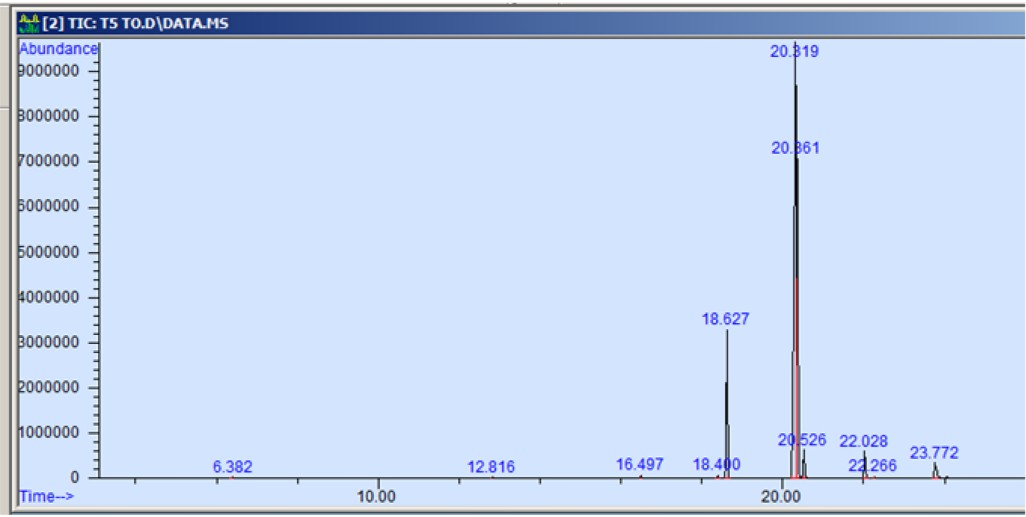

**Figure 3** GC-MS chromatogram of AN 01.

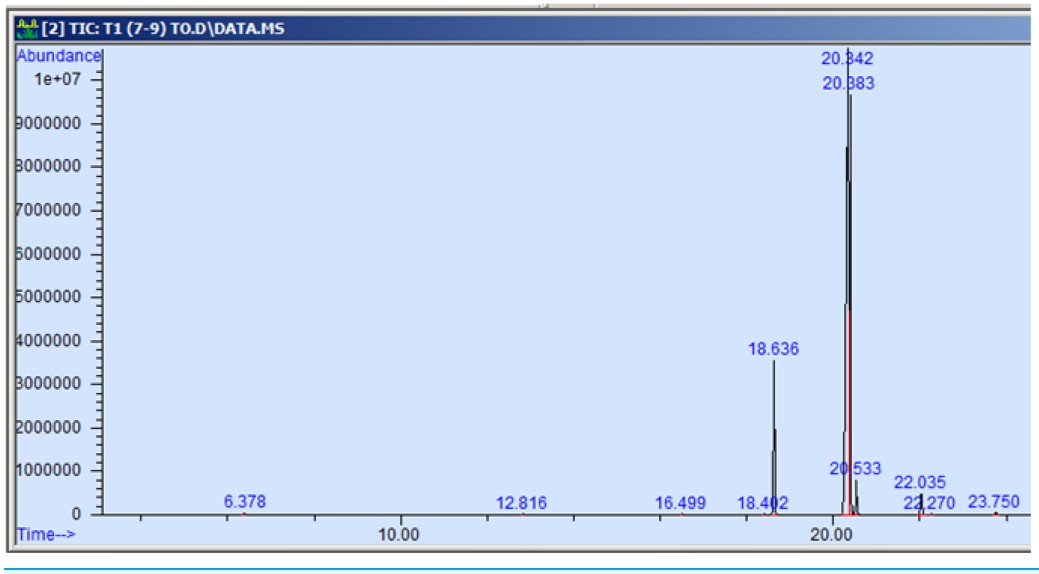

**Figure 4** GC-MS chromatogram of AN 20.

*sativa* L., HPLC analysis was conducted to determine the maximum accumulation of thymoquinone in the released variety of the plant. Prior to thymoquinone quantification, a standard curve was constructed using a stock solution of 1 mg ml$^{-1}$, generating a linear calibration curve with a concentration range of 0.05 to 10 μg ml$^{-1}$. The average coefficient of determination ($R^2$) was found to be 0.998. The method's validation included assessments of linearity, accuracy, robustness, precision, and data selectivity. The resulting chromatograms are illustrated in Fig. 2. The percentage composition varied among the varieties, with Ajmer Nigella-20 recording the maximum percentage composition (0.20 ± 0.07%), and Azad Kalonji reporting the minimum (0.11 ± 0.17%). Phytochemicals produced by a specific plant type could be the cause of fluctuations in yield, leading to
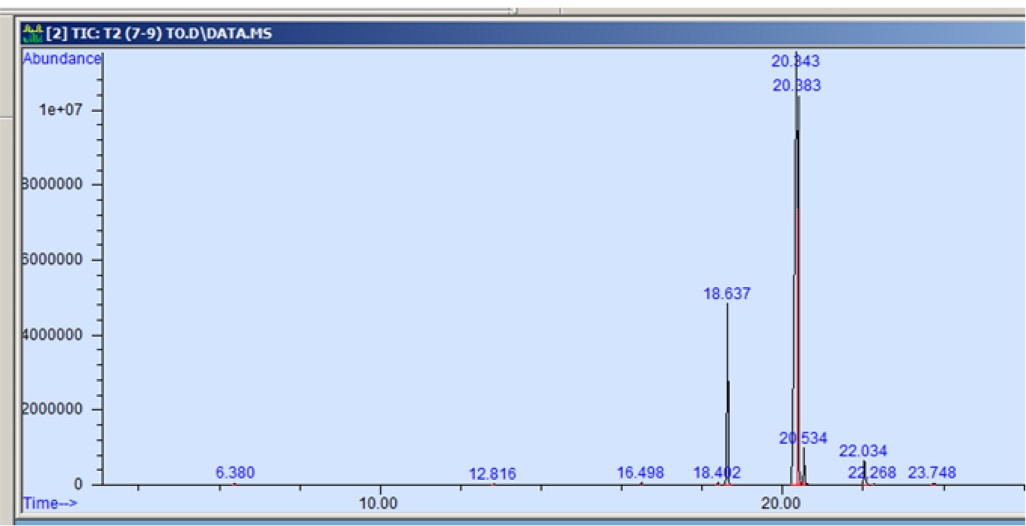

**Figure 5** GC-MS chromatogram of Pant Krishna.

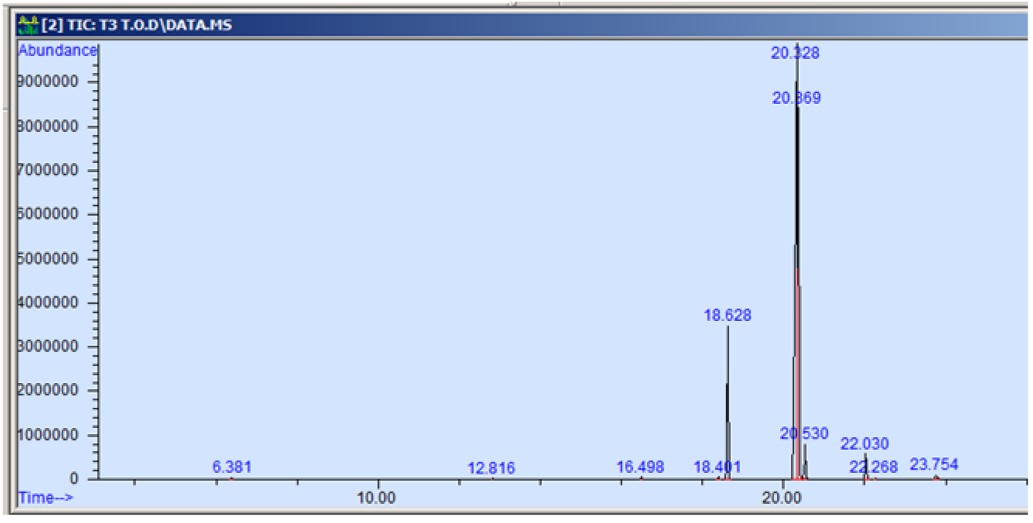

**Figure 6** GC-MS chromatogram of Azad Kalonji.

corresponding changes in the percentage yield. This suggests that Ajmer Nigella-20 is the prominent source of thymoquinone among the studied varieties. The observed variation in thymoquinone quantity is likely linked to the varietal genetic makeup, responding significantly to the given environment and involving physiological and biochemical phenomena that influence the levels of phytochemicals in the crop plant variety.

The distinct quantities of thymoquinone in different varieties imply varied thymoquinone expression, highlighting a connection between percentage yield and phytochemical ingredient makeup. According to research, as the yield increases, so does the percentage composition. Therefore, the percentage yield can be considered a relevant factor for the bioactive ingredients included in the extract.

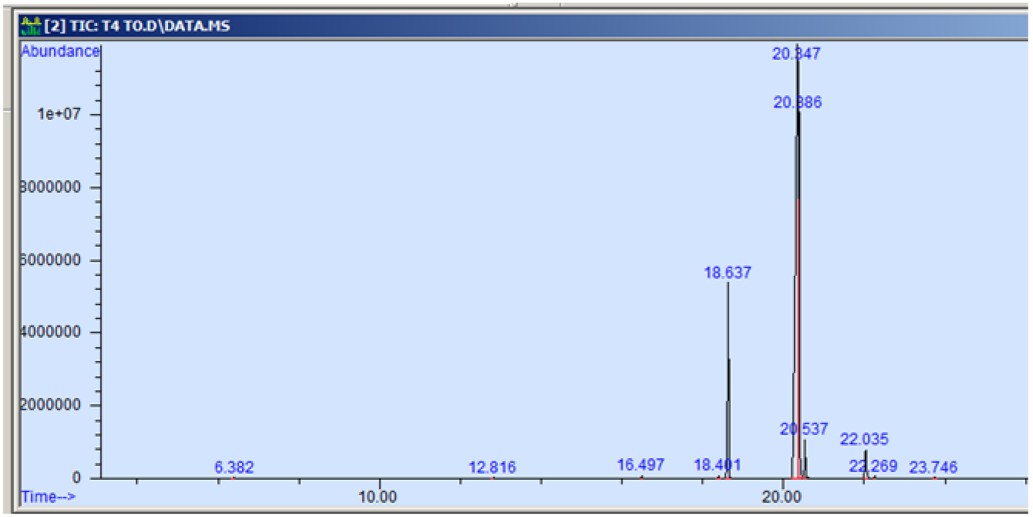

**Figure 7 GC-MS chromatogram of NDBC 10.**  

**Table 3 Phytoconstituents identified by GC-MS from *Nigella sativa* L.**

| Particulars | Mol. weight | Mol. formula | RT | Peak area % |
|---|---|---|---|---|
| p-cymene | 134.22 | C10H14 | 6.383 | 0.115 |
| Longifolene | 204.35 | C15H24 | 12.816 | 0.055 |
| Myristic acid | 242.39 | C15H30O2 | 16.495 | 0.071 |
| 9-hexadecenoic acid, methyl ester | 254.41 | C16H30O2 | 18.401 | 0.058 |
| Hexadecanoic acid, methyl ester | 270.45 | C17H34O2 | 18.639 | 10.374 |
| 9,12-octadecadienoic acid, methyl ester | 280.44 | C18H32O2 | 20.347 | 58.875 |
| 11-octadecenoic acid, methyl ester | 296.50 | C19H36O2 | 20.387 | 25.916 |
| Octadecanoic acid, methyl ester | 298.50 | C19H38O2 | 20.545 | 2.149 |
| Cis-11,14- eicosadienoic acid | 308.50 | C20H306O2 | 22.041 | 1.982 |
| Eicosanoic acid | 312.53 | C20h40O2 | 22.266 | 0.098 |
| 9-octadecenoic acid | 282.46 | C18H34O2 | 23.788 | 0.307 |

The GC-MS profiling of n-hexane extract from varieties of *Nigella sativa* L., revealed the presence of several peaks (Figs. 3–7). The chromatogram peaks were interpreted using the GC-MS library's spectrum of known components database, and Table 3 lists the identified compounds along with their peak area, retention time, molecular formula, and molecular weight. The GC-MS profiling showed the presence of 11 major compounds in the studied varieties, with p-cymene, longifolene, and myristic acid identified as the major chemical compounds present in the oil. These results are consistent with *Khalid & Shedeed (2016)*, where GC-MS analysis of *Nigella sativa L.* showed the existence of sixteen different compounds, with p-cymene, α-thujene, and γ-terpinene identified as major compounds. p-Cymene, also known as p-isopropyl toluene, is a naturally occurring alkyl-substituted aromatic chemical found in essential oils of various medicinal and aromatic plants, as well as several edible plants. Studies have revealed that p-cymene has antioxidant, anti-

inflammatory, antiparasitic, antidiabetic, antiviral, anticancer, antibacterial, and antifungal effects (*Harish et al., 2023*). Additionally, p-cymene has been associated with analgesic, antinociceptive, immunomodulatory, vasorelaxant, and neuroprotective properties (*Verruck et al., 2019*). Myristic acid, another major compound identified, plays a direct role in post-translational protein modifications and mechanisms that regulate essential metabolic processes in the human body (*Legrand & Rioux, 2015*). Modest intake of myristic acid has been associated with increased levels of long-chain omega-3 fatty acids in plasma phospholipids, potentially improving cardiovascular health markers in humans (*Dabadie et al., 2005*). This discussion highlights that the oil from *Nigella sativa* L., has potential applications not only as an edible oil but also due to its pharmaceutical properties.

Table 3 presents the concentration of total phenolic compounds in the studied varieties, assessed using the Folin–Ciocalteau procedure and quantified as gallic acid equivalents. The results indicate that Ajmer nigella-20 had the highest concentration of total phenolic compounds ($31.85 \pm 0.97$ mg GAE/g seed) among the Nigella varieties, while the lowest was observed in NDBC-10 ($30.18 \pm 0.65$ mg GAE g$^{-1}$ seed). These findings align with those of *Saxena et al. (2017)*, who also determined the overall concentration of phenolic compounds in this plant, with an average phenol concentration of 164 mg GAE ml$^{-1}$ in nigella genotypes. Regarding total flavonoids content, the highest was recorded in Ajmer Nigella-20 ($7.31 \pm 0.11$ mg QE g$^{-1}$), while the lowest was in NDBC-10 ($7.31 \pm 0.11$ mg QE g$^{-1}$). This finding is in line with earlier research by *Ahmad et al. (2014)*, where a methanolic extract of *Nigella sativa L.* seeds reported $1.40 \pm 0.29$ mg g$^{-1}$ of total flavonoids. It also corresponds with reports by *Saxena et al. (2017)*, who stated a flavonoid content of 818 mg QE ml$^{-1}$.

The total antioxidant activity was highest in Ajmer Nigella-20 ($76.18 \pm 1.78\%$) and lowest in Pant Krishna ($72.18 \pm 1.95\%$), as measured using the DPPH free radical test, indicating varying levels of antioxidants in the studied varieties (Table 1). The results highlight the high antioxidant activity of Nigella seed oil, consistent with findings in the report by *Gupta et al. (2021)*, where the highest antioxidant activity was reported in the ethanolic extract of the flowering bud. The observed differences in thymoquinone levels among the studied varieties were accompanied by variations in total antioxidant activity, suggesting a connection between thymoquinone quantity and total antioxidant activity. Furthermore, the antioxidant potential of nigella varieties in the DPPH test showed a linear proportional relationship with their total phenolic components. Antioxidant activity increased in direct proportion to polyphenol content, indicating a positive linear association between percent antioxidant activity and total phenolic compounds. The presence of phenolic and flavonoid content is known to enhance antioxidant action. Thymoquinone, carvacrol, anethole, and 4-terpineol, all found in Nigella essential and fixed oil, exhibit significant radical scavenging capacity (*Badary et al., 2003*).

## CONCLUSION

Variations among the studied parameters, including thymoquinone, total phenol content, total flavonoid content, and antioxidant activity, were influenced by nigella varieties.

The findings suggest that the variety Ajmer Nigella-20 would be the better choice among the released varieties for the extraction of thymoquinone and its bioactive compounds. The high levels of total phenols, flavonoids, and antioxidant activity in Ajmer Nigella-20 indicate its potential use in pharmaceutical products to improve health standards. This study provides valuable information on the therapeutic efficacy of the drug, as well as the identification, standardization, and quality control of medicinal plants. The research establishes that different varieties exhibit varying levels of phytochemicals, with Ajmer Nigella-20 showing a greater influence on thymoquinone, total phenols, flavonoids, and a certain group of antioxidants. Based on our analysis, we conclude that seeds of *Nigella sativa* L., contain a variety of beneficial chemical compounds. The presence of these bioactive chemicals validates the traditional use of seeds for treating various ailments by practitioners.

## ACKNOWLEDGEMENTS

The authors are thankful to the Head and Dr. Suresha G., Department of Biochemistry, ICAR-SBI, Coimbatore for their assistance in HPLC analysis.

### Funding

This work was supported by the Horticultural College and Research Institute (HC&RI), Tamil Nadu Agricultural University, Coimbatore, Tamil Nadu, India. The funders had no role in study design, data collection and analysis, decision to publish, or preparation of the manuscript.

### Grant Disclosures

The following grant information was disclosed by the authors:
Horticultural College and Research Institute (HC&RI).
Tamil Nadu Agricultural University.

### Competing Interests

The authors declare that they have no competing interests.

### Author Contributions

- Ravi Y conceived and designed the experiments, performed the experiments, prepared figures and/or tables, authored or reviewed drafts of the article, and approved the final draft.
- Irene Vethamoni Periyanadar conceived and designed the experiments, performed the experiments, prepared figures and/or tables, and approved the final draft.
- Shailendra Nath Saxena conceived and designed the experiments, performed the experiments, prepared figures and/or tables, and approved the final draft.
- Raveendran Muthurajan conceived and designed the experiments, analyzed the data, prepared figures and/or tables, and approved the final draft.

- Velmurugan Sundararajan conceived and designed the experiments, analyzed the data, prepared figures and/or tables, and approved the final draft.
- Santhanakrishnan Vichangal Pridiuldi conceived and designed the experiments, analyzed the data, prepared figures and/or tables, and approved the final draft.
- Sumer Singh Meena conceived and designed the experiments, analyzed the data, prepared figures and/or tables, and approved the final draft.
- Ashoka Narayana Naik conceived and designed the experiments, prepared figures and/or tables, authored or reviewed drafts of the article, and approved the final draft.
- C. B. Harisha conceived and designed the experiments, prepared figures and/or tables, authored or reviewed drafts of the article, and approved the final draft.
- Honnappa Asangi conceived and designed the experiments, prepared figures and/or tables, authored or reviewed drafts of the article, and approved the final draft.
- Sharda Choudhary conceived and designed the experiments, prepared figures and/or tables, authored or reviewed drafts of the article, and approved the final draft.
- Ravindra Singh conceived and designed the experiments, prepared figures and/or tables, authored or reviewed drafts of the article, and approved the final draft.
- Yallappa Dengeru conceived and designed the experiments, analyzed the data, prepared figures and/or tables, authored or reviewed drafts of the article, and approved the final draft.
- Kavan Kumar V conceived and designed the experiments, prepared figures and/or tables, authored or reviewed drafts of the article, and approved the final draft.
- Narottam Kumar Meena conceived and designed the experiments, prepared figures and/or tables, authored or reviewed drafts of the article, and approved the final draft.
- Ram Swaroop Meena conceived and designed the experiments, prepared figures and/or tables, authored or reviewed drafts of the article, and approved the final draft.
- Arvind Kumar Verma conceived and designed the experiments, prepared figures and/or tables, authored or reviewed drafts of the article, and approved the final draft.

### Data Availability

All data have been submitted.

The raw measurements are available in the Supplementary Files 1 and 2.

The supplementary/raw data shows the triplicated data used for the analysis.

### Supplemental Information

Supplemental information for this article can be found online at http://dx.doi.org/10.7717/peerj.17177#supplemental-information.

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
