# Peer review of "Identification, validation and quantification of thymoquinone in conjunction with assessment of bioactive possessions and GC-MS profiling of pharmaceutically valuable crop Nigella (Nigella sativa L.) varieties"

_PeerJ, doi:10.7717/peerj.17177_

## Round 0.1 · original submission · Major Revisions

Dear Dr. Ravi

Thank you for your submission to PeerJ.

It is my opinion as the Academic Editor for your article - Identification, validation and quantification of thymoquinone in conjunction with assessment of bioactive possessions and GC-MS profiling of pharmaceutically valuable crop Nigella (Nigella sativa L.) varieties - that it requires a number of Major and minor changes.

You are therefore advised to carefully go through all the comments and suggestions in order to critically revise the manuscript. You should particularly place utmost emphasis in addressing the comments relating to Materials and Methods section. Moreover, you need to ensure that all the sections of the manuscript are coherent with each, keeping in mind the study objectives and findings. Furthermore, I feel that discussion section needs to be improved.

It is pertinent to mention that your revised manuscript will be evaluated again in order to ensure that you have carefully considered each and every comment raised by the reviewers and the editor.

Hope to receive the revised manuscript in due course.

**Language Note:** PeerJ staff have identified that the English language needs to be improved. When you prepare your next revision, please either (i) have a colleague who is proficient in English and familiar with the subject matter review your manuscript, or (ii) contact a professional editing service to review your manuscript. PeerJ can provide language editing services - you can contact us at copyediting@peerj.com for pricing (be sure to provide your manuscript number and title). – PeerJ Staff

Reviewer 1 ·

Basic reporting

Iam really confused the objective is method development, which I dint see any of it. The TLC was used but not explained the purpose of its usage in results and discussion.

Experimental design

Line92-Are they self seeds or open pollinated seeds? How many replications was used
Line95-machine model no. ?
Line 98- I'm not sure if this is a practical method of determining the bioactive chemicals in oil, particularly at 65 degrees, when the compounds' activities are reduced. If the writers could offer evidence that this isn't the case. I would recommend using the cold press technique to analyse the bioactive component in more detail.

Validity of the findings

I DON’T UNDERSTAND in materials and methods it is about oil extraction, but data are expressed in seed ? Authors should be clear oil, seed meal or whole seeds are completely different and they tell a different story

Additional comments

Comments
Line 65- rephrase sentence
Line92-Are they self seeds or open pollinated seeds? How many replications was used
Line95-machine model no. ?
Line 98- I'm not sure if this is a practical method of determining the bioactive chemicals in oil, particularly at 65 degrees, when the compounds' activities are reduced. If the writers could offer evidence that this isn't the case. I would recommend using the cold press technique to analyse the bioactive component in more detail.
Line 105-The authors did not mention how they visualized the plates.
Line 123-what percentage of methanol was used?
Line 143-151- rephrase the sentences its difficult to understand
Line 153- I DON’T UNDERSTAND in materials and methods it is about oil extraction, but data are expressed in seed ? Authors should be clear oil, seed meal or whole seeds are completely different and they tell a different story.
Line 161-172 pls rephrase sentences its difficult to understand.
Line 184- why reference on turmeric is given?
Iam really confused the objective is method development, which I dint see any of it. The TLC was used but not explained the purpose of its usage in results and discussion.

Reviewer 2 ·

Basic reporting

satisfactory.
The manuscript presents good information on "Identification, validation and quantification of
thymoquinone, other bioactive compounds and GC-MS analysis of Nigella", The language is easy to understand and proper context is provided.

Experimental design

No comment

Validity of the findings

No comment

Additional comments

Some of the suggestions for improving the manuscript are as follows.
List of Comments
1. #61. Other chemicals
Comment: List the other chemicals used
2. #59. Mention the chemical name and formula of the thymoquinone
3. #72. Essential oil extraction methods are a significant factor in the properties of oils
Comment: Rewrite the sentence for better understanding
4. #79. Replace medicines with metabolite/chemical constituent
5. #81. The objective of the study was to
6. #73.Solvent extraction is the modern method of extraction that has a low selectivity and 74 superior method compared to other methods
7. Comment: the whole sentence need to restructured in a proper manner.
8. #92. Nigella sativa L.,
9. #103. Reference is not in proper format
10. 107. #Sample extracted in n-hexane was 108 spotted along with the standard thymoquinone on the TLC plates and kept chromatograms were 109 developed with the mobile phase comprising n-hexane and ethyl acetate
Comment: rewrite the sentence
11. Provide reference for HPLC instrumentation
12. #125. In methanol, seven calibration curves TQ reference points (0.05, 0.1, 0.2, 1, 2.5, 5, 10 µg/mL) depicted in Fig. 1.
Comment: Rewrite the sentence for better understanding
13. #130. Add reference for GC-MS profiling
14. #131. The chemical composition of seed oil
15. #158. 415nm wavelength not 15nm spectrophotometer
16. #224 and #263 N. sativa L.,
Comment: Nigella sativa L.,

·

Basic reporting

Article title: Identification, validation and quantification of thymoquinone in conjunction with assessment of bioactive possessions and GC-MS profiling of pharmaceutically valuable crop Nigella (Nigella sativa L.) varieties”
Abstract:
Comment 1: The introduction to the abstract is extremely extensive, and it should place more emphasis on the findings and conclusions. For readers to comprehend the procedures and results, the abstract should be self-contained. It is necessary to rewrite the abstract and include the samples
Introduction:
The selection of the topic and title of the manuscript are appropriate and significant concerning the current scenario with special reference to the importance of medicinal plants for human health care and well-being.
Comment 1. Solvent extraction is the modern method of extraction that has a low selectivity and superior methods compared to other methods. Here, the authors need to highlight the other methods of extraction and their drawbacks compared to solvent extraction.
Comment 2. The authors have done GC-MS profiling of the elected varieties but no information has been compiled related to GC-MS. Hence, introductory information needs to be added for a better understanding of the results.
Comment 3. Authors need to highlight the objectives of the study clearly so the readers can benefit and plan further advanced studies in the said area.
Comment 4. Provide a multi-panel figure for Nigella sativa L., varieties.
Comment 5. Grammatical typo in the whole manuscript must be corrected.

Experimental design

Material and Methods:
In material and methods, the planning, design and conduct of the experiment are appropriate and in accordance with the objectives of the study. The authors need to address the following comments;
Comment 1. Provide the source of the manufacturing company or from where the standards and chemicals were purchased to conduct the experiment.
Comment 2. Zhang, Bi & Liu, 2007 the cited reference is appropriate. Correct it.
Comment 3. 415nm spectrophotometer. Is it wavelength or spectrophotometer? Check it again and rectify it.
Comment 4. DPPH radical scavenging activity test:
DPPH is a photosensitive agent do the authors keep the reaction mixture under dark conditions? If so, kindly mention it in the material and methods section.
Comment 5. Cite reference for HPLC and GC-MS profiling published in previous issues from PeerJ journal.
Comment 6. Figures and tables formatting should be improved

Validity of the findings

Results and Discussion:
The results are aptly tabulated and expressed and the units of measurement are properly cited. The discussion is appropriate and supported by the research work done by the previous workers in the similar line in different crops are appreciable.
Comment 1. The discussion on bioactive possessions of results needs to be supported with the latest available references. Put forward to Cite a few of the results published in PeerJ journal.

Additional comments

Conclusion:
The salient findings of the study are summarized appropriately as per the experimental results.
The research manuscript is recommended for acceptance with incorporation of typographical errors and comments suggested in different sections must be done accordingly. The findings may be published for better visibility and dissemination of the generated results.

---

## Round 0.2 · accepted · Accept

Dear Dr. Ravi

Thank you for your submission to PeerJ.

Based on the perusal of the reviewers comments and my own assessment, this is to inform you that your manuscript - Identification, validation and quantification of thymoquinone in conjunction with assessment of bioactive possessions and GC-MS profiling of pharmaceutically valuable crop Nigella (Nigella sativa L.) varieties - has been Accepted for publication.

Congratulations!


This is an editorial acceptance; publication is dependent on authors meeting all journal policies and guidelines.

Reviewer 2 ·

Basic reporting

Satisfactory

Experimental design

Satisfactory

Validity of the findings

Satisfactory